# Visual acuity of *Empoasca onukii* (Hemiptera, Cicadellidae)

**DOI:** 10.3390/insects14040370

**Published:** 2023-04-08

**Authors:** Chang Tan, Xiaoming Cai, Zongxiu Luo, Zhaoqun Li, Chunli Xiu, Zongmao Chen, Lei Bian

**Affiliations:** 1Tea Research Institute, Chinese Academy of Agricultural Science, 9 Meiling South Road, Xihu District, Hangzhou 310008, China; 2Key Laboratory of Tea Biology and Resource Utilization, Ministry of Agriculture, 9 Meiling South Road, Xihu District, Hangzhou 310008, China

**Keywords:** visual ability, *Empoasca onukii*, compound eye, structure, visual acuity

## Abstract

**Simple Summary:**

*Empoasca onukii*, a small hemipteran insect, is a serious pest of tea plants. Visual cues play an important role in habitat localization of adult *E. onukii*. In this study, we found that an acute zone exists in the compound eyes of adult *E. onukii*. A structural trade-off of the compound eye ensures that *E. onukii* has low-resolution vision (approximately 0.14 cycles per degree).

**Abstract:**

*Empoasca onukii* is a common tea plant pest with a preference for the color yellow. Past work has shown that host leaf color is a key cue for habitat location for *E. onukii*. Before studying the effect of foliage shape, size, or texture on habitat localization, it is necessary to determine the visual acuity and effective viewing distance of *E. onukii*. In this study, a combination of 3D microscopy and X-ray microtomography showed that visual acuity did not significantly differ between females and males, but there were significant differences in the visual acuity and optical sensitivity among five regions of *E. onukii*’s compound eyes. The dorsal ommatidia had the highest visual acuity at 0.28 cycles per degree (cpd) but the lowest optical sensitivity (0.02 μm^2^sr), which indicated a trade-off between visual resolution and optical sensitivity for *E. onukii*. The visual acuity determined from the behavioral experiment was 0.14 cpd; *E. onukii* exhibited low-resolution vision and could only distinguish the units in a yellow/red pattern within 30 cm. Therefore, visual acuity contributes to the limited ability of *E. onukii* to distinguish the visual details of a distant target, which might be perceived as a lump of blurred color of intermediate brightness.

## 1. Introduction

Visual cues are important for insects’ ability to search for food, locate hosts, and navigate in flight. The quality of visual information acquired by insects is limited by the visual ability of compound eyes [1]. *Empoasca onukii* is a major pest of tea plants and has a significant preference for the color yellow [2]. In a previous study, we found that foliage color was an important habitat location cue for *E. onukii* [3]. Color is a basic parameter in the visual information of host leaves. Additionally, the shape and texture of leaves affect host localization of herbivorous insects [4]. The visual acuity of insect compound eyes is an important ability that determines whether insects can effectively use these parameters at a long distance. However, the visual acuity and effective viewing range of *E. onukii* have not yet been reported.

The compound eyes of *E. onukii* are apposition eyes, which have tubular structures composed of the cornea, crystalline cones, and photoreceptor cells that transmit light to the rhabdom [5]. Multiple ommatidia cluster the identified images onto the retina, which is made up of several retinular cells [6]. In the insects’ visual system, visual acuity is the parameter that helps them resolve static spatial details. The better the acuity is, the greater the distance at which environmental structures can be used during localization and movement [7]. Moreover, visual acuity can be quantified with methods such as apposition eye analysis [8].

The acuity index is closely related to compound eye structure. Two aspects of compound eye morphology fine-tune visual acuity in insects: interommatidial angle and separation angle [9]. The interommatidial angle, which is the separation angle of the photoreceptors, is determined by the ommatidia density in each region. The higher the density is in the same area, the higher the visual acuity [7]. Therefore, the smaller the interommatidial angle is, the better the visual acuity. In insects for which this information is known, acuity ranges from 0.01 cycles per degree (cpd) (bat fly, *Trichobius frequens*) to 2 cpd (dragonfly, *Anax junius*) [10]. Most insects have low visual acuity compared with human eyes, which have a visual acuity of 60 cpd; low acuity, to some extent, limits the ability to perceive details of the environment or specific patterns. The acceptance angle is the angular width of the region seen by each photoreceptor, the ommatidium, and is related to the ommatidium area. The smaller the ommatidium is, the more it is optically affected by diffraction [11]. However, for most diurnal insects with compound eyes, the upper limit of visual acuity can be estimated by the interommatidial angle [9]. It should be noted that, without understanding the mechanism of visual input after it reaches the visual nerve, the interommatidial and acceptance angles may only indicate physical limitations of the compound eye structure. A recent study showed that some insects possess muscles that can move the retina the distance of a few ommatidia [12]. Therefore, combining visual analysis with behavioral experiments may reveal the responsiveness of insect vision in specific environments [13].

*Empoasca onukii* is an insect that undergoes incomplete metamorphosis. During the progression from nymph to adult, its range of activity increasingly expands, and the shape of its compound eyes also changes from spherical to kidney shaped; this indicates that its visual acuity is constantly changing and may differ in different regions of the compound eye. The measure of visual acuity, expressed as cpd, is the number of black and white stripe pairs that the photoreceptors can identify within an angle and is also represented by the angle of the narrowest black-and-white-stripe pair identified by the photoreceptor. Cycles per degree can be read in degrees (in which case it would be the reverse value); that is, the angular width of the narrowest black-and-white-stripe pair that can be discerned [8]. When the experimental conditions have been carefully selected, visual acuity values of insects measured by anatomical and behavioral methods are highly correlated [10]. Kirwan et al. [14] exploited object taxis to test the spatial resolution of *Euperipatoides rowelli* through a behavioral assay. Similarly, based on the taxis to yellow color, behavioral experiments can be designed to measure the visual acuity of *E. onukii* compound eyes.

In this study, we measured the optical structural parameters of *E. onukii* compound eyes using histological techniques and obtained a theoretical value of their visual sensitivity. A piecewise sine stimulus of yellow and black grating was used as the target in behavioral experiments. Both the maximum distance at which *E. onukii* could recognize the stimulus and their behavioral visual acuity were analyzed based on the behavior of *E. onukii*. Finally, the visual recognition of a yellow/red pattern was simulated to assess the viewing distance of *E. onukii.*

## 2. Materials and Methods

### 2.1. Insects

Adult *E. onukii* were collected from the tea garden of the Tea Research Institute at the Chinese Academy of Agricultural Sciences. Plants of the “Longjing 43” tea cultivar were hydroponically raised to rear insects in a 26 °C constant temperature insect-rearing room. Adults at 3 to 4 days after emergence were selected for experiments.

In total, 8 males and 8 females were used for external 3D structure analysis, 3 females for internal 3D structure analysis, and 65 males and 107 females for behavioral experiments.

### 2.2. External 3D Structures

The external 3D structure of compound eyes was measured using a digital microscope with a VH-Z100UR lens at 500×–700× (Keyence VHX-6000, Japan). The depth resolution of the 3D microscope was 0.45 µm. Because the single compound eye of *E. onukii* does not have uniform symmetry, the optical structural parameters of ommatidia in five different regions (posterior, ventral, central, anterior, and dorsal) were analyzed. First, adult *E. onukii* were each fixed in 4% paraformaldehyde solution for 30 min and then rinsed three times with 0.1 M phosphate-buffered saline (PBS) at pH = 7.0 for 15 min [15]. Before scanning, each sample was fixed with plasticine and placed on the microscope platform. After the orientation of the compound eye was adjusted, the five regions of the compound eye were scanned (Figure 1A). In total, 8 adult male and 8 adult female samples were analyzed.

The Interommatidial angle and the ommatidium diameter were measured using the manufacturer’s software. Each hexagonal ommatidium was divided in three axial directions according to the midline of the opposite side, and the angle corresponding to the radius of curvature between the center points of several adjacent ommatidia was calculated from the side view. The angle was divided by the number of ommatidia (Figure 1B), and the mean value of the three axial angles was regarded as the mean interommatidial angle (Δ*ϕ*) [16]. The diameter of a single ommatidium was the mean distance between the longest diagonal (*D*h) and the shortest edge (*D*v) of the hexagonal ommatidium. Eight ommatidia were selected and measured to calculate the mean diameter (*D*) for each region.

An independent-samples *t*-test was used to analyze the differences of Δ*ϕ* and *D* between females and males. After confirmation that there was no difference in Δ*ϕ* and *D* between the sexes, we pooled the data from both sexes and analyzed the differences in Δ*ϕ* and *D* among the five regions in compound eyes using one-way ANOVA and LSD comparison (SPSS 14.0, Inc., Chicago, IL, USA).

### 2.3. Internal 3D Structures

Based on the above experimental results, the internal structure of three female *E. onukii* compound eyes was observed via a high-resolution X-ray microscopy imaging system (SKYSCAN1272, Bruker, Karlsruhe, Germany). Adult *E. onukii* were fixed in 4% paraformaldehyde solution overnight and rinsed three times with 0.1 M PBS at pH = 7.0 for 15 min each. For secondary fixation, the samples were treated with 1% osmic acid solution for 2 h and rinsed three times with 0.1 M PBS at pH = 7.0 for 15 min each. The samples were dehydrated in a graded series of ethanol for 15 min at different concentrations (30%, 50%, 70%, 80%, 90%, and 95%), treated with 100% ethanol for 20 min, and then stored in fresh 100% ethanol. The dehydrated samples were dried in a critical point dryer (HCP-2, Hitachi, Tokyo, Japan). The dried samples were then scanned in the area of the head using the microscopy imaging system with a source voltage of 50 kV, source current of 55 µA, scanning image pixel size of 1.00 µm, and scanning time of 2 h; 750 images were reorganized by Avizo (2019.1, Thermo Fisher Scientific, Waltham. MA, USA) [17]. The lengths (*L*) of 38 rhabdoms in 5 regions were measured using ImageJ (1.8.0, USA).

The sensitivity (*S*) of an ommatidium was then determined with the following formula:*S* = (π/4)^2^ *D*^2^Δ*ϕ*^2^(*kL*/(2.3+*kL*))(1)
where *S* has the units of μm^2^sr, *D* = diameter of the female ommatidia (μm), and *L* = the length of the photoreceptor (μm). The absorption coefficient, *k* = 0.0067 μm^−1^, was from Warrant and Nilsson [18].

A one-way ANOVA was performed to analyze the differences in the lengths of rhabdoms of the five regions of female *E. onukii* compound eyes, and LSD comparison was used to determine differences in the data of each group.

### 2.4. Behavioral Experiments

For apposition eyes, visual resolution is structurally affected by the interommatidial angle, and the resolution determines the ability of the compound eyes to distinguish spatial details, which behaviorally manifests as the ability to identify light and dark gratings at a distance. Kirwan et al. [14] used white and black grating as the piecewise sine stimulus in their vision resolution study. Because of the taxis of *E. onukii* to yellow targets, yellow and black grating was designed as the piecewise sine stimulus in the behavioral experiment in this study.

The experiment was performed in a dark chamber, with a cylindrical reaction area of 15 cm diameter and 5 cm height (Figure 2A). Below the reaction area was a luminosity plate of 815 µw/cm^−2^, which was derived from the light intensity during the peak period when *E. onukii* were active in clear weather. The background on the side of the reaction chamber was gray, and the target area was composed of yellow and black gratings of equal width. The piecewise sine stimulus consisted of a dark (negative) half-period sine flanked on each side by a light (positive) half-period sine of half the amplitude (Figure 2B,C). It was also a prerequisite for our experiments that the stimuli were isoluminant against the gray background if averaged across their whole width. *Empoasca onukii* adults will exhibit taxis toward high-brightness targets in their visual field; therefore, the purpose of isoluminance was to avoid adults choosing the target because of a difference in brightness and instead respond by recognizing the grating. Before each test, the reaction chamber was cleaned with ethanol and dried.

Before each behavioral test, an individual *E. onukii* was dark-adapted for 10 min. The individual was then released at the farthest point from the grating (release point *a*, Figure 2D). Each *E. onukii* was allowed to randomly move until after reaching a specific point (such as identification point *b*, Figure 2B); a directional movement trend started until the yellow zone was contacted, which was considered the end of the test. The directional movement trend was usually completed rapidly, i.e., within 30 seconds. Each behavioral test was limited to 10 min. If the test exceeded 10 min, *E. onukii* were deemed to have no response to the target. Behavioral experiments were repeated for a total of 172 adults, and the blank control was repeated 15 times without stimulus.

The movement trajectory of *E. onukii* was recorded by a high-speed camera (Revealer 2F01M, China). The recording started at starting point *a* and ended when *E. onukii* reached identification point *b*. The video analysis software Tracker (V6.0.8, USA) was used to calibrate the two-dimensional coordinate system, and the position of *E. onukii* during the search process was marked every 10 frames to determine the individual action trajectory. The angle between each point *b* and the width of the yellow area was measured (Figure 2B), which was regarded as the behavioral interommatidial angle of each *E. onukii*.

The frequency distribution of the interommatidial angle values was constructed based on the results of the behavioral tests; then, a nonlinear regression (log normal, 3 parameter) was performed in SigmaPlot (V11.0, Systat Software Inc., San Jose, CA, USA). The peak value of the curve was regarded as the behavioral interommatidial angle (Δ*ϕ*’) at which *E. onukii* detected the stimulus grating.

### 2.5. Viewing Distance Assessment

The AcuityView package in R (V4.2.0) was used to simulate *E. onukii* recognition of static images to assess effective viewing distance [19]. Three parameters, interommatidial angle (Δ*ϕ*’), target picture, and test distance, were entered in the R program to simulate *E. onukii* visual acuity at the test distances. The target picture was a color card (20 × 25 cm) with a checkerboard pattern of small red and yellow squares (5 × 5 cm), and this pattern has been reported to be attractive to *E. onukii* adults in the field [3]. The interommatidial angle was determined from behavioral experiments, where the target picture was a yellow/red patterned card with a 256 × 256 resolution, and the test distance was the length between the compound eye and the target. This approach provides a theoretical value of effective viewing distance of image details that are difficult to distinguish.

## 3. Results

### 3.1. Histological Visual Acuity

Histological results showed no significant differences between sexes in the interommatidial angle (Δ*ϕ*) or ommatidia diameter (*D*) in different regions of adult *E. onukii* compound eyes (Table 1).

Data from both sexes were pooled and analyzed. Among the five regions, the dorsal interommatidial angle was the smallest (Δ*ϕ* = 3.57° ± 0.13, visual acuity = 0.28 cpd) while the posterior ommatidia had the largest interommatidial angle (Δ*ϕ* = 8.52 ± 0.46°, visual acuity = 0.12 cpd); both significantly differed from those of the other three regions (*F* = 38.84, *df* = 4, *p* < 0.05; A). The ommatidia diameter in the dorsal region (*D* = 7.80 ± 0.30 µm) was the smallest and significantly differed from those in the posterior, ventral, and anterior regions (*F* = 6.43, *df* = 4, *p* < 0.05; Figure 3B).

### 3.2. Optical Sensitivity

Rhabdoms in different regions are closely connected to the lens (Figure 4A–C). There were significant differences in the lengths of rhabdom of the five regions of female *E. onukii* compound eyes (*F* = 19.02, *df* = 4, *p* < 0.05; Figure 4D). The shortest rhabdom (*L* = 68.76 ± 0.76 µm) in the ventral region of the compound eyes had a small degree of bending, and the longest rhabdom was in the posterior region (*L* = 93.21 ± 4.28 µm).

There were significant differences in the optical sensitivities (*S*) of rhabdoms among five regions of adult *E. onukii* compound eyes (*F* = 509.2, *df* = 4, *p* < 0.05; Figure 5), which indicated that the light absorption capacity differed among regions in the same compound eye. The sensitivity of the posterior region was the highest (*S* = 0.22 μm^2^sr), whereas that of the dorsal region was the lowest (*S* = 0.02 μm^2^sr). The phenomenon of high resolution and optical sensitivity in the dorsal region indicates that there is a structural and functional trade-off in the compound eye, and optical sensitivity is sacrificed to improve resolution in the dorsal region of *E. onukii* compound eyes.

### 3.3. Behavioral Visual Acuity

In the behavioral experiment without stimulus grating, *E. onukii* behavior trajectories were nondirectional, curving, and spiral (Figure 6A). In comparison, *E. onukii* performed random movement after release and then presented stable trends of moving toward a 5 × 5 cm wide piecewise sine stimulus. During the random process, *E. onukii* exhibited spiral flight, short distance jumping, and stable crawling behavior. After an individual detected the yellow stimulus, it moved toward the yellow area with a slightly curved trajectory (Figure 6B). Each *E. onukii* that exhibited no response remained in the same position and barely moved.

Of the 172 total trending behaviors, 149 effective responses were recorded with the camera. Among the 149 effective responses to the stimulus, 79% of the interommatidial angles were between 6–8°, and the smallest angle reached 4.5°; these values were within the range of histological structure measurements. Figure 7 shows that the peak value of the curve in the frequency distribution diagram was the behavioral interommatidial angle (Δ*ϕ*’= 7.3°, *R*^2^ = 0.83), which had a reciprocal value of 0.14 cpd (behavioral visual acuity).

### 3.4. Viewing Distance Assessment

According to the behavioral visual acuity of 0.14 cpd, adult *E. onukii* could visually perceive blurred details of a 20 × 25 cm yellow/red pattern at 0–30 cm (Figure 8A–D). Beyond a distance of 30 cm, differences within the pattern could not be distinguished, and the pattern appeared as a single blurred color lump of intermediate brightness beyond 50 cm (Figure 8D, E). The sharpness of the pattern gradually decreased as the distance between the individual *E. onukii* and the target increased.

## 4. Discussion

There is an acute zone in the dorsal region of adult *E. onukii* compound eyes that is characterized by large ommatidia diameters and small interommatidial angles [20]. The acute zone is related to behavioral habits, such as tracking mates or avoiding predation [13]. For example, the honeybee drone (*Apis mellifera*) uses its dorsal eye, which is the acute zone, to detect and approach the queen from behind and below during mating behavior [21]. However, visual cues are not important in the process of mate localization in *E. onukii* [22]; therefore, *E. onukii* might use dorsal ommatidia to observe the upper environment and detect potential predators.

Because of the limited compound eye area of *E. onukii*, the same region cannot simultaneously have high resolution and optical sensitivity. Visual heterogeneity and the trade-off between resolution and sensitivity are necessary for behavioral requirements of small insects [23]. In general, larger insects have better visual acuity [17,24], but the “bright areas” and “love spots” formed by compound eye specialization can satisfy the trade-off of the distribution of visual resolution and optical sensitivity among different areas of the compound eye [25,26], which is a fine-tuned mechanism of compound eyes shaped by long-term evolution. In predatory insects such as *Holcocephala fusca* (Δ*ϕ* = 0.28°), which have a small body size, the nearly planar region of the anterior part of the compound eye provides high-resolution recognition of flying prey, and their vision in specialized regions is comparable to that of the dragonfly (*Anax junius*), which has compound eyes that are several times larger (Δ*ϕ* = 0.24°) [27,28]. As the compound eye grows at each molt, new ommatidia appear at the anterior region and the existing ommatidium move posteriorly [29], which may be the reason why the posterior ommatidia of *E. onukii* have long rhabdom and high optical sensitivity. The posterior ommatidia of *E. onukii* are more likely to combine with other regional ommatidia to provide light intensity differences in the whole field of view rather than discrimination, which facilitates adjustment of travel direction [12].

The visual acuity of insects is closely related to their flight ability and range of movement. *E. onukii* adults are usually in a resting state, and leaf tenderness can be identified by foliage-reflected light intensity so that the most suitable living environment can be selected [6]. The single flight distance of *E. onukii* adults is very limited, and they mainly rely on spiral flight, curved crawling on the plane, and short hops through the tea plants. To date, no evidence of long-distance migration has been reported. Therefore, the visual ability of *E. onukii* compound eyes (Δ*ϕ* = 7.3°) can meet their survival needs. Similar to psyllids (*Hemiptera: Psylloidea: Aphalaridae*) (Δ*ϕ* = 6.3°), the low-resolution vision of *E. onukii* limits their ability to perceive details and recognize hosts or conspecific species at long distances [15]. For most long-distance, fast-flying insects, such as bees, flies, and butterflies, the Δ*ϕ* range is between 1–3°, and they can see longer distances. Predatory invertebrates such as dragonflies, mantises, and spiders have a minimum Δ*ϕ* of less than 1°, and they can see smaller prey travelling at higher flight speeds [9].

In our previous study, the yellow/red patterned sticky cards (20 × 25 cm) could be used to trap *E. onukii* adults in the field [3]. While the surface area of the yellow in the bicolor traps was half that of the yellow sticky cards, similar numbers of trap-collected *E. onukii* were obtained in the field, likely because of their strong attraction to yellow. At a distance beyond 30 cm, sticky cards with 5 × 5 cm yellow and red squares would appear yellow (Figure 8), they would likely appear less bright and merge into yellow with intermediate brightness, and thus are similarly attractive to the insects [30]. However, the reason why bicolor sticky and yellow sticky cards showed similar trapping effects in the field needs further research, especially with regard to the effective attraction range of the two types of sticky cards and the flight ability of *E. onukii*.

## 5. Conclusions

There is an acute zone in the compound eyes of adult *E. onukii*. To obtain visual resolution, *E. onukii* sacrifices some optical sensitivity in the dorsal ommatidia. The structural trade-off of the compound eye ensures that *E. onukii* has a low-resolution vision of approximately 0.14 cpd; in other words, this is only sufficient to recognize a pattern (20 × 25 cm) of yellow and red square units (5 × 5 cm) within 30 cm. Beyond 30 cm, *E. onukii* might perceive the bicolor patterns as yellow of intermediate brightness.

## Figures and Tables

**Figure 1 insects-14-00370-f001:**
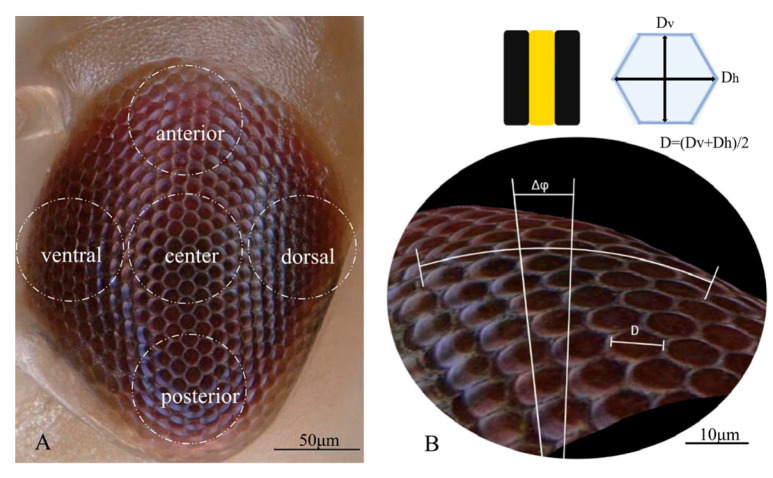
Interommatidial angle (Δ*ϕ*) and ommatidia diameter (*D*) measurements of the compound eye. (**A**) The compound eye is divided into five regions: posterior, ventral, central, anterior, and dorsal. (**B**) Side view of the adjacent ommatidia interommatidial angle and diameter. The hexagonal ommatidia with six adjacent ommatidia were used to measure the radius of curvature, and the corresponding angle (Δ*ϕ*) was calculated between two ommatidia. The diameter of a single ommatidium was the mean distance between the longest diagonal (*D*h) and the shortest edge (*D*v) of the hexagonal ommatidium. The ommatidia diameter (*D*) was the mean of eight ommatidia in each region.

**Figure 2 insects-14-00370-f002:**
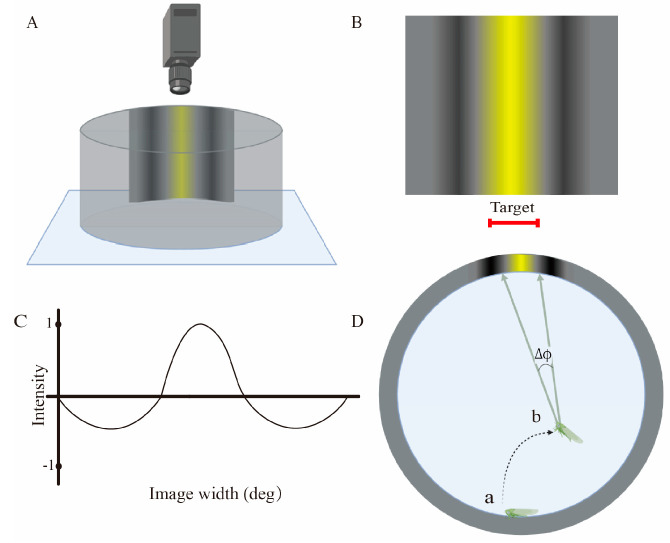
Diagrammatic drawing of the behavioral experiment setup. (**A**) Side view of the experimental setup, with the high-speed camera above. In the middle is the behavioral reaction setup (15 cm diameter, 5 cm height). A yellow–black grating with 5 × 5 cm patches was placed on the side wall, and the overall brightness of the grating was the same as that of the gray background on the side wall. Below is the luminescent background plate with a light intensity of 815 µw/cm^−2^. (**B,C**) The yellow–black grating and its brightness correspond to the piecewise sine stimulus. (**D**) Top view of the behavioral experiment setup. *Empoasca onukii* were released at point *a* and conducted spontaneous search behavior to point *b*; during this time, if *E. onukii* exhibited a recognition response to the yellow–black grating, the angle Δ*ϕ* between the individual and the target grating was measured.

**Figure 3 insects-14-00370-f003:**
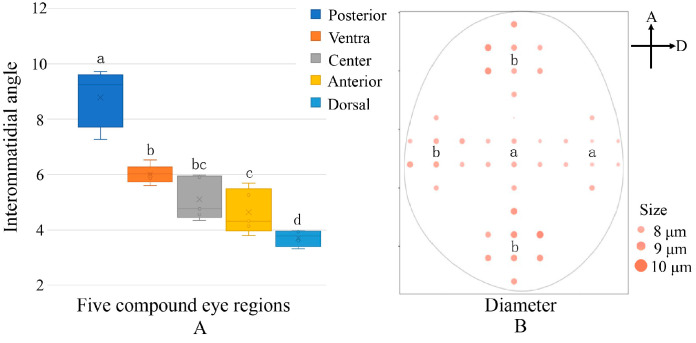
Interommatidial angle (Δ*ϕ*) and ommatidia diameter (*D*) of *Empoasca onukii* compound eyes (mean ± SE). (**A**) Box plot of Δ*ϕ* in five different regions. (**B**) Ommatidia diameter distribution in different regions of the same compound eye; eight ommatidia were measured in each region. The different letters indicate significant differences (*p* < 0.05, one-way ANOVA).

**Figure 4 insects-14-00370-f004:**
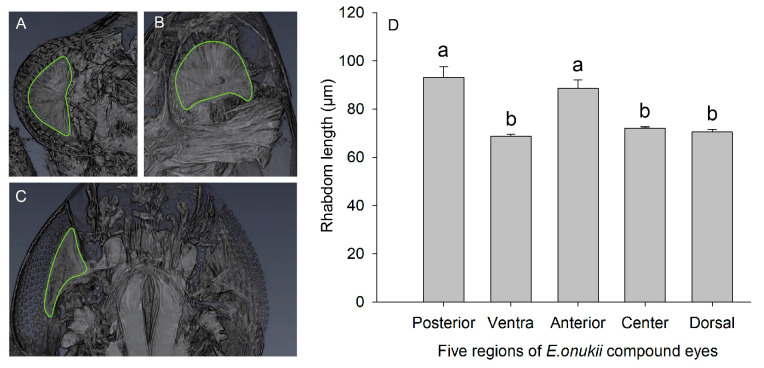
Internal structure and the lengths of rhabdom in five regions of the female compound eye (mean ± SE). (**A**) From top to bottom: dorsal, central, and ventral. (**B**) From left to right: ventral, central, and dorsal. (**C**) From top to bottom: anterior, central, and posterior. (**D**) Rhabdom lengths in five different regions. The marked area in A–C is the rhabdom. The different lowercase letters in D indicate significant differences (*p* < 0.05, one-way ANOVA).

**Figure 5 insects-14-00370-f005:**
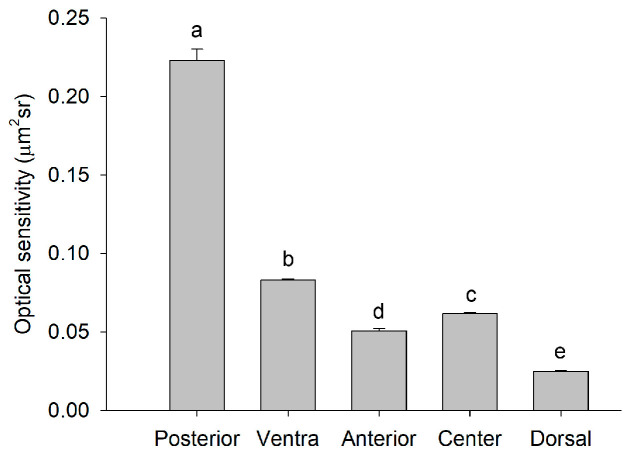
The optical sensitivity (*S*) in the posterior, ventral, anterior, central, and dorsal regions of female *Empoasca onukii* compound eyes (mean ± SE). The *S* value was calculated from Equation (1). The different letters indicate significant differences (*p* < 0.05, one-way ANOVA).

**Figure 6 insects-14-00370-f006:**
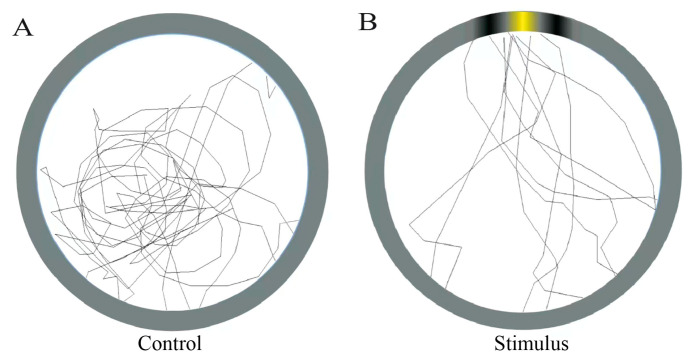
Movement behavior trajectories of *Empoasca onukii* in behavioral experiments. (**A**) Blank control, *E. onukii* search trajectories in the absence of stimulus and (**B**) *E. onukii* search trajectories in the presence of stimulus.

**Figure 7 insects-14-00370-f007:**
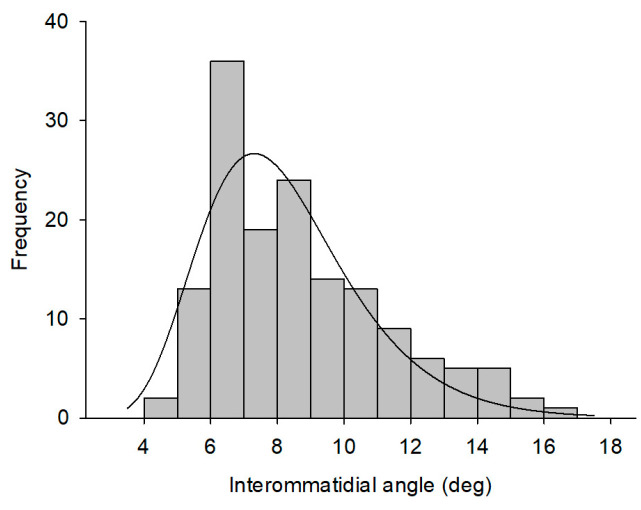
Frequency distribution diagram of interommatidial angles from the behavioral experiment. The x-axis is the frequency distribution of interommatidial angles with the same interval. The fitting curve (log normal, 3 parameter) was obtained using SigmaPlot (V11.0).

**Figure 8 insects-14-00370-f008:**
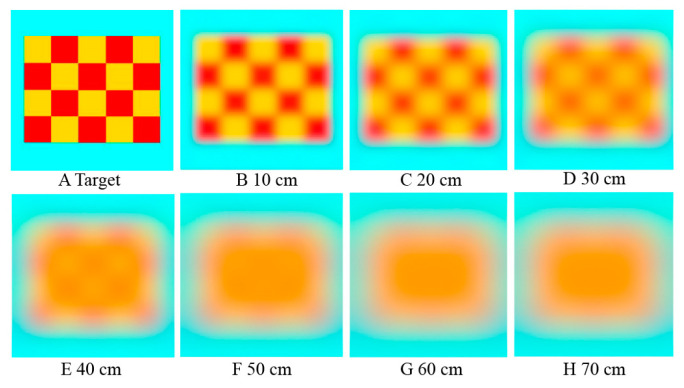
Simulations of the viewing distance of *Empoasca onukii*. (**A**) Target yellow/red pattern with 256 × 256 resolution. (**B**–**H**) Visual simulations of *E. onukii* viewing a 20 × 25 cm yellow/red pattern at different distances with a visual acuity of 0.14 cpd was conducted using the AcuityView package in R (V4.2.0).

**Table 1 insects-14-00370-t001:** Interommatidial angle (Δ*ϕ*) and ommatidia diameter (*D*) of male and female *Empoasca onukii* compound eyes (mean ± SE).

	Dorsal	Ventral	Posterior	Anterior	Center
Interommatidial angle (Δ*ϕ*)					
Female	3.46 ± 0.08	5.83 ± 0.12	8.83 ± 0.78	4.72 ± 0.49	5.07 ± 0.42
Male	3.68 ± 0.26	5.85 ± 0.44	8.21 ± 0.59	4.58 ± 0.43	4.93 ± 0.53
*t*	0.80	0.07	0.63	0.21	0.21
*p*	0.47	0.99	0.56	0.84	0.84
Ommatidia diameter (*D*)					
Female	7.97 ± 0.19	8.82 ± 0.21	8.79 ± 0.54	8.88 ± 0.19	8.11 ± 0.22
Male	7.64 ± 0.33	8.98 ± 0.19	8.99 ± 0.59	9.02 ± 0.13	8.30 ± 0.11
*t*	0.89	0.59	0.25	0.62	0.76
*p*	0.43	0.59	0.81	0.57	0.49

Significant differences between males and females were determined with an independent-samples *t*-test (*p* < 0.05).

## Data Availability

The data presented in this study are openly available in [FigShare] at [10.6084/m9.figshare.22578250].

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
