# Peer review of "Visual acuity of Empoasca onukii (Hemiptera, Cicadellidae)"

_insects, 2023, doi:10.3390/insects14040370_

Round 1
Reviewer 1 Report (Previous Reviewer 1)
Figures improved in clarity. I think authors did very poor efforts in putting context in their article. Reading the 2 previews version I did not understand why authors keep insisting on why E. onukii could somehow differentiate the red and gold colour pattern. They simply never wrote that in their previous experiment authors compared different sizes of red and yellow colour patterns and obtained different results with different sizes of red squares. For example L52-54: “… we found that sticky cards with a chequerboard pattern of small red and yellow squares (5×5 cm) sticky cards with yellow/red patterns did also attracts lots of E. onukii adults, but did not decrease E. onukii trapping efficiency”. Did not decrease the trapping efficiency compared to what? This should be clearly defined by the authors. Unfortunately, authors did not have direct control over different red square patterns and no red patterns, which limits the interpretation. So again, I would restrict the link from this work to their previous work to only a sentence or two in the discussion.
L56-57. Did you measure the light reflected light intensity? If not, please remove the sentence. The reason why you say that they might perceive the different colours or patterns is that you observed a difference in behaviour from your previous study. But you do not know if it is from being able to perceive different colours or because their visual acuity is too bad, or both.
L468: are likely to be apparent.
L477: Likely appear
Figure 3: I don’t understand why that figure moved to the Material and methods instead of staying in the results. Authors did not comment on that in their reply. If there is no good reason, I think it is best that this figure stays in the result section.
Figure 6: What is n=10?? I thought you had 172 behaviours recorded. That figure can be removed, it does not add to the manuscript.
Author Response
Responses to the comments of Reviewer #1
Figures improved in clarity. I think authors did very poor efforts in putting context in their article. Reading the 2 previews version I did not understand why authors keep insisting on why E. onukii could somehow differentiate the red and gold colour pattern. They simply never wrote that in their previous experiment authors compared different sizes of red and yellow colour patterns and obtained different results with different sizes of red squares. For example L52-54: “… we found that sticky cards with a chequerboard pattern of small red and yellow squares (5×5 cm) sticky cards with yellow/red patterns did also attracts lots of E. onukii adults, but did not decrease E. onukii trapping efficiency”. Did not decrease the trapping efficiency compared to what? This should be clearly defined by the authors. Unfortunately, authors did not have direct control over different red square patterns and no red patterns, which limits the interpretation. So again, I would restrict the link from this work to their previous work to only a sentence or two in the discussion.
Response: We apologize for any confusion associated with this. Our original intention was to briefly introduce the previous research as the reason for our current research. Obviously, this will confuse the readers.
So, as your suggestions, we removed the relevant content describing the ability of E. onukii to distinguish colors and focused on the visual acuity in this revision of manuscript, including simple summary (lines 10-17), abstract (lines 18-24), the first paragraph of the introduction (lines 38-53), and the last paragraph of the discussion (lines 371-379).
- L56-57. Did you measure the light reflected light intensity? If not, please remove the sentence. The reason why you say that they might perceive the different colours or patterns is that you observed a difference in behaviour from your previous study. But you do not know if it is from being able to perceive different colours or because their visual acuity is too bad, or both.
Response: Thank you for the comment. This sentence has been removed accordingly and rewritten. (lines 51-53).
- L468: are likely to be apparent.
Response: Thank you for the comment. This sentence has been removed after revision. (lines 374-379).
- L477: Likely appear
Response: Thank you for the comment. This has been revised as requested. (lines 383).
- Figure 3: I don’t understand why that figure moved to the Material and methods instead of staying in the results. Authors did not comment on that in their reply. If there is no good reason, I think it is best that this figure stays in the result section.
Response: In the section 2.4, we first mentioned the figure 3, so we moved it to this section. As your request, this figure was moved back to the result after revision. (lines 326-330).
- Figure 6: What is n=10?? I thought you had 172 behaviours recorded. That figure can be removed, it does not add to the manuscript.
Response: Thank you for the comment. We guess you were referring to Figure 7 in the original manuscript. In figure 7, 10 samples were used to visualize the trajectory, so as to visually present the movement trajectory of E. onukii adults.
According to your suggestion, we removed the numbers in the figure. (lines 301-306, Figure 6)

Reviewer 2 Report (New Reviewer)
Tan el al combine anatomy analyses and behavior experiments to determine the visual acuity of a leafhopper species, Empoasca onukii, which is a major pest for tea production. The anatomy results suggests vary visual acuity in different locations of the eye, and a trade-off between acuity and light sensitivity. The visual acuity measured via behavior experiments shows a low value than what one expects from anatomy. In addition, there is no evidence for significant differences in visual acuity between sexes. Here are some comments need to be addressed:
- “cpd” is not a standard unit and should be spelt out the first it appears. Currently it wasn’t done until line 75.
- What’s the depth resolution of the 3D microscope ? This will affect the calculation of the radius of curvature – in other words, what’s the uncertainty of this measurement ?
- Line 108, “biological replicates” ? You mean samples ?
- Line 115, this is not clear to me. An easy way to explain is to add a diagram.
- Line 147, these is no “A” in your equation
- Line 167, please explain more why you need isoluminance. In fact, I’m confused what you mean by this since black has no luminance by definition.
- How did you generate this stimulus experimentally ?
- The name ‘leafhopper’ first appeared in line 181, then another 2 more times in the rest of the manuscript. It adds unnecessary confusion.
- Line 187, do you only rotate by 90 degrees left or right ? Then there are in total 3 discrete locations, no? This doesn’t solve the chemical trail problem.
- How many trials end up as “no response” ? What’s the animal trajectories for these trials ?
- Figure 7, why many trials don’t start at the boundary ?
-
Author Response
Responses to the comments of Reviewer #2
Tan el al combine anatomy analyses and behavior experiments to determine the visual acuity of a leafhopper species, Empoasca onukii, which is a major pest for tea production. The anatomy results suggests vary visual acuity in different locations of the eye, and a trade-off between acuity and light sensitivity. The visual acuity measured via behavior experiments shows a low value than what one expects from anatomy. In addition, there is no evidence for significant differences in visual acuity between sexes.
Response: Thank you for your approval and suggestions. We have revised the manuscript according to your comments. We hope the revised manuscript can fill the accepted requirements.
Here are some comments need to be addressed:
- “cpd” is not a standard unit and should be spelt out the first it appears. Currently it wasn’t done until line 75.
Response: Thank you for your comment, we spelt out the full name of “cpd” where it first appeared. (line 69)
- What’s the depth resolution of the 3D microscope? This will affect the calculation of the radius of curvature – in other words, what’s the uncertainty of this measurement ?
Response: The depth resolution of the 3D microscope is 0.45 µm, which has been added after revision. (lines 113-114)
- Line 108, “biological replicates”? You mean samples?
Response: We apologize for the confusion associated with this. The ‘biological replicates’ was replaced with ‘samples’ after revision. (line 121)
- Line 115, this is not clear to me. An easy way to explain is to add a diagram.
Response: Thank you for your comment. We added a diagram describing the method for calculating the diameter of an ommatidium in Figure 1. (lines 128-129, lines 137-145 Figure 1)
- Line 147, these is no “A” in your equation
Response: We apologize for this typographical error. The ‘A’ has been replaced with ‘D’. (line 162)
- Line 167, please explain more why you need iso-luminance. In fact, I’m confused what you mean by this since black has no luminance by definition.
Response: E. onukii adults will present taxis to high-brightness targets in their visual field, so the purpose of iso-luminance between the stimulus (black-yellow-black grating) and the background is to avoid E. onukii adults choosing the target due to brightness difference, but to conduct behavioral response by recognizing the grating.
In this way, since E. onukii adults can only distinguish the yellow and black texture in the grating and show a taxis response within the effective viewing distance, behavioral experiments can be used to estimate the visual acuity of the compound eyes of E. onukii adults.
After revision, the necessary information has been added. (lines 183-186)
- How did you generate this stimulus experimentally?
Response: Based on the method of Kirwan et al. (2018), we used yellow and black grating as the piecewise sine stimulus. The picture containing stimulus and background was generated by a software edited through Python QT, which was provided by Prof. Qing Yao in Zhejiang Sci-Tech University.
After revision, this information has been added in the Acknowledgments. (lines 406-407)
- The name ‘leafhopper’ first appeared in line 181, then another 2 more times in the rest of the manuscript. It adds unnecessary confusion.
Response: We apologize for any confusion associated with this. We have replaced the word ‘leafhopper’ with ‘E. onukii’. (lines 199, 323-324 and 361)
- Line 187, do you only rotate by 90 degrees left or right? Then there are in total 3 discrete locations, no? This doesn’t solve the chemical trail problem.
Response: We apologize for this confusion, here we made a typographical error.
Measures to avoid the effect of chemical trails on the behavior of E. onukii have been mentioned above: ‘Before each test, the reaction chamber was cleaned with ethanol and dried’ (line 186). So, this sentence was deleted after revision. (lines 205-207)
- How many trials end up as “no response”? What’s the animal trajectories for these trials ?
Response: In total, there were 23 trials end up as “no response”, which has been mentioned in the result 3.3 (line 308-309). After the behavioral test began, individual of “no response” remained in the same position and barely moved.
After revision, the information has been added accordingly. (lines 298-299)
- Figure 7, why many trials don’t start at the boundary?
Response: After being released at point a, some E. onukii crawled along the side wall and then jumped to the bottom of the cylindrical reaction room. This process was very long and apparently E. onukii did not see the target of the stimulus. Some trajectories of this process we forgot to label. After revision, we supplemented it with the video analysis software Tracker (V6.0.8). (lines 302-306 Figure 6)

This manuscript is a resubmission of an earlier submission. The following is a list of the peer review reports and author responses from that submission.
Round 1
Reviewer 1 Report
Review insects-2069469: “Visual acuity and ability to distinguish bicolor patterns of Empoasca onukii (Hemiptera, Cicadellidae)”
The article is generally well written and add new insight on visual acuity in an underrepresented group of insects. I think the experiments were well done, and data are strong. However, I have a few concerns that authors should address. I think authors made assumptions that are not supported in the literature on for their primary hypothesis. I do not understand why authors keep thinking that E. onukii can perceive differences in colours because similar number of insects were caught on monochromatic yellow or yellow and red squares. If anything, this experiment support that the insect cannot. Authors seems to think that E. onukii can discriminate between yellow and red, but there is no evidence in the literature from the same insect family for this (developed further). Also, authors take a very anthropogenic view, that likely does not reflect what the insect sees. The light intensity was measured in Lux, a measure based specifically on the human eye sensitivity, not on the absolute measure of light intensity (such as number of photon).
L41: Ref 1, please clarify if E. onukii and E. vitis are the same species. This ref should be placed at the end of the previous sentence L39.
L45-46: I really don’t understand that statement. If you had no differences in insect catch between monochromatic yellow vs. 5x5 yellow and red square, why do you assume the insect can differentiate both patterns?? On the opposite, it would indicate to me that if there is no difference in catch between both that this insect is enabled to differentiate between yellow and yellow + red or can differentiate but it does not affects its behaviour.
This is a very anthropic view, different to what insect can perceive. Most insects also detect UV light, that can be reflected on paint or the support but were not previously measured. On one hand, Cicadellidae seems to only have one opsin for long wavelength absorption (Guignard et al 2022). This is supported by the only one species in this family where electroretinogram measurement showed one of each UV, blue and green photoreceptor (van der kooi et al. 2021). E. onukii likely follow the same pattern. That means that E. onukii might not discriminate yellow from red as they look the same colour, but of different intensity. However, if one colour is brighter (e.g reflect more light) than the other one, both colours activate the same photoreceptor (likely since they are both long wavelengths). In that case, E. onukii might not be able to discriminate between the red and yellow colours presented in the previous experiment. Since the absorption spectrum and intensity were not measured in the previous manuscript, both possibilities are possible.
In the article cited (“Design and selection of trap colour for capture of the tea leafhopper, Empoasca vitis, by orthogonal optimization”) there were no control on the different intensity of the light reflected not the different wavelength (especially UV). So it is unclear (from that reference only) if the gold colour tested was the most attractive because of the wavelength or because of its intensity.
Neither “Sticky Card for Empoasca Onukii with Bicolor Patterns Captures Less 363 Beneficial Arthropods in Tea Gardens” nor “Design and selection of trap colour for capture of the tea leafhopper, Empoasca vitis, by orthogonal optimization” articles tested true black, and white as “insect white” (e.g. excite all the photoreceptors) is different for insects and human, especially since insects detect UVs. So, none of these experiments can support the fact that Empoasca Onukii. Based on these studies, no assumption can be made on colour discrimination of Empoasca Onukii, only preference and field attraction.
L46: Do you mean the red colour or the square pattern?
L142-144: Another example of the anthropogenic view of the authors. First, you do not know that the “yellow” light is monochromatic as you have not measured its spectral absorption. For example, UV could be present, and the insect might respond to these more than the yellow.
Second, the lux was scaled on the sensitivity of the human eye. Different light of the same intensity might give you different lux measurements if they emit in a different part of the spectrum. Again, if a lot of UVs are present, they won’t be detected with a lux meter. I recommend the authors to use a spectrometer and measure the number of photons / cm2 to measure the light intensity. This instrument should also be used to show the absorbance spectrum of the light they used in their behavioural experiment.
L146: “The overall brightness of the grating was equal to that of the gray back-ground…” from a human perspective, since it was measured with a Lux meter and not a spectrometer but might be differently perceived by the insect. Now, you don’t have the right measurement for that statement.
L162: unclear, please reformulate: “a directional trend of moving until it reached the yellow area “. Does that mean the insect walk randomly from point “a” to “b” (Figure 2) ?
L292-304: You must be cautious with this paragraph. The RGB colour panner corresponds to the red, green, and blue photoreceptor that (trichromatic) human express. But only three opsins and three corresponding photoreceptors (UV, blue, green) were found in Cicadellidae (Guignard 2022, van der koi 2021), so it is safe to assume that this is the case in that E. onukii. It is possible that one opsin can be used for two different photoreceptors with tuning mechanisms (e.g different retinal, screening or filtering pigments). But so far, none were described in this family (and order?) of insects. It is therefore possible that the yellow and red colours look like the same colour to E. onukii, but at different intensities as the green opsin (likely the only opsin able to absorb yellow and red photons) will be activated. I think it is very likely that you have no differences between yellow and yellow + red square in the attraction of this insect simply because it might not have the ability to discriminate both as different colours (simply different intensity, if the red is still visible). A similar phenomenon occurs in deuteranope people, missing the red opsin and only having a green opsin to discriminate long wavelengths (e.g. green, yellow, orange, red, all look similar, but intensities vary). So, at a longer distance, the sticky card with 5x5 yellow and “red” square would be a yellow and less bright yellow square that merges into an intermediate brightness of yellow, rather than a new colour. This would likely not affect the attraction of E. onukii if from a distance it perceives a monochromatic yellow vs. a less bright yellow and could explain why you have no difference in trap catch based on your study and what was previously described in the literature.
L305-306: Italicise E. onukii
L323-336: This paragraph is not relevant to this study, suppress it. In addition, the measurements were done in lux again.
Figure 1a: I found this figure confusing, especially with the orientation it is presented. It seems like the eyes belong to an insect in a vertical position, as suggested by the posterior/anterior and ventral/dorsal legend. If yes, then inserting the picture of an insect with the same orientation as the eye would help the reader to quickly understand. I would also change the name “frontal” by “centre” throughout the manuscript as “frontal” means the front part.
Figure 3b: The legend “F” and “D” is confusing. Does it mean “frontal” and “dorsal”? If yes, why are they perpendicular? If this figure is the same orientation as Figure 1a, shouldn’t the top be “A” for anterior and “D” for dorsal? Please make this figure the same orientation as figure 1 so we know in which direction the insect is looking at.
Are the different points within the same area different individuals or geographically different areas of measure? In this figure, it seems to be the different geographical areas of measure. If it is not, I recommend the author make a similar figure (boxplot or average ± STD) similar to Fig 3A.
Also, what is the difference between the two “Size” legend?
Figure 4D: Not readable. Please also orient it the based-on figure 1A.
Figure 5: Change for: “Bar plots of optical sensitivity in the posterior, ventral, anterior, frontal, and dorsal regions of E. onukii compound eyes”.
Author Response
The article is generally well written and add new insight on visual acuity in an underrepresented group of insects. I think the experiments were well done, and data are strong. However, I have a few concerns that authors should address. I think authors made assumptions that are not supported in the literature on for their primary hypothesis. I do not understand why authors keep thinking that E. onukii can perceive differences in colours because similar number of insects were caught on monochromatic yellow or yellow and red squares. If anything, this experiment support that the insect cannot. Authors seems to think that E. onukii can discriminate between yellow and red, but there is no evidence in the literature from the same insect family for this (developed further). Also, authors take a very anthropogenic view, that likely does not reflect what the insect sees. The light intensity was measured in Lux, a measure based specifically on the human eye sensitivity, not on the absolute measure of light intensity (such as number of photon).

Reviewer 2 Report
The authors are investigating the visual sensitivity of E. onukii, a well-known tea plant pest in the context of previous studies they have made. They previously worked on improving pest control solutions to avoid non-target insects (including pollinators) to be trapped. They found that red/yellow sticky cards were a good solution to this problem but wanted to understand why. Hence, this study addresses this question.
The study is overall interesting and sound. It addresses a question that your previous work has raised and can be useful for species comparisons and development of pest control for other taxa. Overall, the draft needs to be improved regarding : the summary/introduction that should explain how this work came to be, the explanation of the methods, all the figures have to be improved (quality, police, image resolution), the statistics used in which context, the results aren't sufficiently self-explanatory and the discussion could use more comparison to vision work in insect. English should be improved also.
Additional comments can be found in the file 'Review 08.12.22'.

Author Response
The authors are investigating the visual sensitivity of E. onukii, a well-known tea plant pest in the context of previous studies they have made. They previously worked on improving pest control solutions to avoid non-target insects (including pollinators) to be trapped. They found that red/yellow sticky cards were a good solution to this problem but wanted to understand why. Hence, this study addresses this question.
Point 1: The study is overall interesting and sound. It addresses a question that your previous work has raised and can be useful for species comparisons and development of pest control for other taxa. Overall, the draft needs to be improved regarding: the summary/introduction that should explain how this work came to be, the explanation of the methods, all the figures have to be improved (quality, police, image resolution), the statistics used in which context, the results aren't sufficiently self-explanatory and the discussion could use more comparison to vision work in insect. English should be improved also.

Round 2
Author Response
Dear Editor,
Thank you very much for providing an opportunity to resubmit our revised manuscript. We appreciate your positive and constructive comments and suggestions as well as those of the reviewers regarding our manuscript. Please find enclosed our manuscript “Visual acuity of Empoasca onukii,” which we would like to be considered for publication in Insects. All of the authors have contributed significantly, agree to the submission of this paper, and declare that we have no conflict of interest.
Empoasca onukii is a serious tea pest that preferentially inhabit the tender leaves of tea plants. Here, we founded that E. onukii has low-resolution vision (approximately 0.14 cpd) and can visually recognize the details of a target in the environment at a certain distance.
We hereby certify that this paper consists of original, unpublished work that is not under consideration for publication elsewhere.
Best Regards.
Yours Sincerely,
Chang Tan
Responses to the comments of Reviewer #1
Major point
1.The manuscript improved in quality after the revision. Authors did a good job making the manuscript more interesting and easier to read, especially in their M&M, results and discussion. However, I still think authors focus too much on discrimination in colour patterns when their manuscript shows evidence for visual acuity and contrast sensitivity. It is good that the authors discuss about it (L377-386), but they should remove every other part that relate to the ability of E. onukii to discriminate colours, as there is no evidence for it. I understood that authors really want to pile up on their previous work and want to understand why the bicolour pattern was not more attractive/repellent, but results of this study are largely insufficient to answer this question. Ability to distinguish colour depend on visual acuity, but also neural connection, opsin expression, photoreceptor distribution, spectral sensitivity and more information that were not studied here.
Response: Thank you for your suggestion, we have made a major revision to our manuscript, including deemphasizing the content describing the ability of E. onukii to distinguish colors, and concentrating on the visual acuity of the E. onukii. (line 392-405)
- Title should be changed. There is no evidence of the ability to distinguish bicolor patterns. Authors should focus on visual acuity.
Response: As suggested by the academic editor, the title of the article has been changed to ‘Visual acuity of Empoasca onukii’, and the ability of the E. onukii to distinguish colors has been weakened (line 2).
Minor points
- L31: You have no proof in your study that “light sensitivity” affect the behaviour of the insect, remove it.
Response: Thank you for your suggestion, we removed this part after the revision (line 31).
- L55: “The bicolor sticky card was made of basic…..”.
Response: Thank you for the suggestion. This sentence has been revised as requested. The revised sentence is ‘The bicolor sticky card was made of basic monochromatic yellow and red units were 5×5 cm in size’. (line 60)
3.L56: “Yellow and red have different light intensities…”. I guess you mean “The yellow and red units of the bicolor sticky trap reflect different light intensity…”, or “The photoreceptor sensitivity might be different for the yellow and red wavelength”.
Response: We are sorry for the confusion and have made the suggested changes. The revised sentence is ‘The yellow and red units of the bicolor sticky trap reflect different light intensity under the same lighting conditions; therefore, we speculate that E. onukii may have adequate visual acuity to detect the small yellow units of the bicolor patterns in relation to distance(line 60-63).’
- L57: Again, I don’t understand why autors speculate that “E. onukii distinguishes the details of the bicolor patterns in relation to distance” if there is no difference in trap catches! And their results are in line with that assumption.
Response: We apologize for any confusion. This paragraph was rewritten with deemphasizing the content describing the ability of E. onukii to distinguish colors (line 63-64)
.
- L65: You don’t know for sure the E. onukii is the same, but you assume so.
Response: Thank you for the suggestion. This confused part has been removed after revision (line 65).
- L68-69: Again, remove “therefore, the ability of E. onukii to discriminate between bicolor patterns is not based on spectral sensitivity.” As you have no evidence even in your previous study that the insect can distinguish bicolor patterns.
Response: Thank you for the suggestion. This part has been removed after revision (line 68-69).
- L74: Remove the “. “ and lower cap “Dividing”
Response: Thank you for the suggestion. This part has been removed after revision (line 74).
- L192:227: I advise to restructure so the text follow the order of Figure 2 (A, B, C, D).
Response: Thank you for the suggestion. The figure has been modified (Figure 2).
- L323:329: Please use the same range as in your figure, or adapt the figure. For example, we see a skewed distribution between 6.2 and 7.9 degree in Fig7, not at 6.5 degree. Same for the most interommatidial angle most frequently visited.
Response: Thanks to your suggestion, we shortened the spacing of the horizontal coordinate data and revised the figure 7. After the logarithmically curve fitting it was determined to be 7.3 degree (line 350,353).
- L428-432: Rephrase. The insect won’t perceived two distinct colour at close range and different intensity of yellow at long range. From your data, it is likely to distinguish different intensities of “yellow” at close range, that would merge into an intermediate intensity at longer range.
Response: Thank you for the suggestion. This sentence has been revised as requested. The revised sentence is ‘At a longer distance, sticky cards with 5x5 “yellow” and “red” squares would appear “yellow”, they would just appear less bright and merge into yellow with intermediate brightness. Therefore, the reason why bicolor sticky and yellow sticky cards showed simi-lar trapping effects in the field experiment needs further research, including the effective attraction range of two types of sticky cards and flight ability of E. onukii(line 401-405).’

Reviewer 2 Report
Dear authors,
The concerns I had about the draft were, in some cases, not adressed properly. The english is still poor in certain paragraphs (new), all the figures haven't been modified, the explanations given to my questions aren't (always) convincing. The conclusions that you made aren't supported by the current results. There were improvements, however your manuscript still cannot be accepted as it is unfortunately.
Best,

Author Response
Dear Editor,
Thank you very much for providing an opportunity to resubmit our revised manuscript. We appreciate your positive and constructive comments and suggestions as well as those of the reviewers regarding our manuscript. Please find enclosed our manuscript “Visual acuity of Empoasca onukii,” which we would like to be considered for publication in Insects. All of the authors have contributed significantly, agree to the submission of this paper, and declare that we have no conflict of interest.
Empoasca onukii is a serious tea pest that preferentially inhabit the tender leaves of tea plants. Here, we founded that E. onukii has low-resolution vision (approximately 0.14 cpd) and can visually recognize the details of a target in the environment at a certain distance.
We hereby certify that this paper consists of original, unpublished work that is not under consideration for publication elsewhere.
Best Regards.
Yours Sincerely,
Chang Tan
Responses to the comments of Reviewer #2
Answers to reviewers should all be inside your cover letter, and thus, we should not have to look and search for each of your changes inside your manuscript. Moreover, the lines you give in your reviewed manuscript do not match with the answers. You didn’t articulate your arguments and answered properly to most of my questions which renders this review system inefficient.
Response: Sorry for our previous unsatisfactory answer, since we didn't find your comments in the review system until the deadline, so many revisions were made in a hurry and very weak. I apologize again, and in this revision we will answer some of the previous questions together. As the comments from the academic editor and other reviewers, they suggested us to deemphasize the content describing the ability of E. onukii to distinguish colors and focused on the visual acuity of the leafhopper. We strongly endorse these suggestions, and the confusions you have mentioned also were caused by these contents. It is incorrect for us to make assumptions when much of the evidence is insufficient (the ability of E. onukii to distinguish colors).
1.L.23 ‘patterns as yellow of intermediate’
Response: Thank you for the comment. The sentence has been revised accordingly (line 23).
- L.53 ‘did also attracts’
Response: Thank you for the comment. The sentence has been revised accordingly (line 58).
- L.56 If yellow or yellow/red patterns equally attract your target insect, your H0 should be they cannot distinguish the difference between them.
Response: Thank you for the suggestion. The original data were not sufficient to clarify whether E. onukii have the ability to distinguish between yellow and red, so the sentence has been revised (line 57-63).
- L.60-83 Is this copy/paste from a text book? Summarise.
Response: We are sorry for the confusion and have removed the inappropriate statement (line 60-83).
- L. 68 ‘the ability of E. onukii to discriminate between bicolor patterns is not based on spectral sensitivity.’ You still haven’t demonstrated this ability, you cannot say that here. This does not constitute an answer to my first question and does not respond to is properly.
Response: Thank you for the suggestion, we have made a major revision to the manuscript, weakening the content description of the E. onukii ability to distinguish colors and concentrating on the visual acuity of the E. onukii (line 57-63).
- L.70-80 This paragraph serve no function and is not connected at all to the rest of your introduction, it doesn’t support your assumptions – remove or rewrite. This does not constitute an answer to my first question and does not respond to is properly.
Response: We are sorry for the confusion and have removed the inappropriate statement (line 70-80).
- L.119 ‘more about’ more than what?
Response: We apologize for any confusion. This part has been removed after revision (line119).
- L.122-124 This phrase makes no sense - ‘According to the preference of adult E. onukii for
yellow, sticky cards with yellow and black grating was were designed as the piecewise sine
stimuli. – rephrase
Response: We apologize for our typographical error, which has now been corrected.The revised sentence is ‘According to the preference of adult E. onukii for yellow, color cards with yellow and black grating was designed as the target. Both the maximum distance at which E. onukii could recognize a target and their visual acuity were was analyzed uncovered using by behavioral experiments, and E. onukii visual acuity was evaluated based on their behavior(line 134-139).’
- L.124-126 ‘Both the maximum distance at which E. onukii could recognize a target and their visual acuity were was analyzed uncovered using by behavioral experiments, and E. onukii visual acuity 125 was evaluated based on their behavior.’
Response: Thanks to your suggestions. We have modified this paragraph according to your suggestion (line 134-139).
- L.130-133 This does not constitute an answer to my first question and does not respond to is properly.
Response: We apologize for any confusion. We have added to the selection and number of insects. Adults from 3 to 4 days after emergence were selected for histological and behavioral experiments. A total of 8 males and 8 females compound eyes were photographed for histological experiment, 3 females for micro-CT, and 38 males and 65 females for behavioral experiments.
- L.142 ? Where is the reference asked in my previous review?
Response: We apologize for any confusion. The reference has been added (line 91).
- L. 155-156 ‘One-way ANOVA revealed significant differences in visual acuity and diameters of the five regions of adult E. onukii compound eyes,’ – results should be in the result section not in methods. Methods should only describe what you used and how, not the result.
Response: We apologize for any confusion. We have rewritten the whole paragraph to clarify our intended meaning (line 273-276).
- L. 180-182 ‘One-way ANOVA revealed significant differences in the lengths of rhabdom of the five regions of adult 181 E. onukii compound eyes,’ – same here, should be in results not in methods.
Response: We apologize for any confusion. This sentence has been moved to the section of results (line 292-296).
- L.184 This does not constitute an answer to my first question and does not respond to is
properly.
Response: We apologize for any confusion. We have rewritten the whole paragraph, and this sentence has been deleted(line 184).
- L. 196 You changed Lux (which is for human vision) to ‘μw/cm-2’ can you give an explanation why using this measure for insects? And in your context.
Response: We apologize for any confusion. The choice of units comes from the references, (Land, M. F. and D.-E. Nilsson. Animal eyes. Oxford University Press, 2012). P29 “A radiometric system using energy units (watts = joules per second) is essentially the same as the photon number system, except that the units are watts (or microwatts) rather than photons per second. The conversion factor is Einstein’s equation, given earlier: E = hν = hc / λ, where h is Planck’s constant, ν is frequency, c is the speed of light (3.10 8 m.s −1) and λ is wavelength (in metres). For photons in the yellow-green region of the spectrum (555 nm) this works out as 3.6 × 10 –19 joules. Thus one watt of yellow-green light is equivalent to about 2.8 × 10 18 photons per second.”
- L.199-200 This does not constitute an answer to my first question and does not respond to
is properly. – you have to explain what are the difference between your experiment and the
one you cite. If you made changes explain what supports these changes.
Response: We apologize for any confusion. We used the same sample pre-treatment method as in Ref., but the instrument used in our experiment was different from that in Ref., in which the authors used X-ray microtomography. The 3D super-depth microscope was used in our experiment to obtain 3D images of insects.
- L. 223 ‘When the experiment was repeated, the target was arbitrarily rotated 90 deg to avoid the effect of chemical trails and the position of the experimenter’ this is not a sufficient explanation and the phrase is unfinished.
Response: We apologize for any confusion. We have modified this expression. The revised sentence is ‘When the experiment was repeated, the behavioral reaction setup was arbitrarily rotated 90 deg to avoid the effect of chemical trails and the position of the experimenter (line 233-235).’
- L. 226 ‘with a male to female ratio of 1:2’ why this ratio?
Response: We apologize for any confusion. Compared to females, males are more active, so they are used less (line 150).
- L. 242-244 ‘Enter the optimum interommatidial angle, target picture (Figure 8A) and test distance in the R program to generate a visual simulation of E. onukii at the test distance.’ –
not English, please rewrite.
Response: We apologize for any confusion. We have modified this expression.
‘The following three parameters, including visual acuity, target picture (Figure 8A) and test distance, were entered in the R program to generate the E. onukii visual simulation at the test distance. Visual acuity is the reciprocal of the interommatidial angle determined from behavioral experiments; The target picture is yellow/red patterned card with 256×256 resolution; The test distance is the length between the compound eye and the target (line 252-257).’
- L. 256 - 258 Space needed between the table and the text paragraphs (up/down)
Response: Thanks to your suggestions. We have fixed the format (line 270).
- Figure 3 is still not publishable as it is – y coordinates are too small, bars in the boxplots too
thin, legend – unable to read. – this was not revised accordingly.
Response: Thank you for the suggestion. This figure has been modified (Figure 3).
- Figure 5 – still unreadable – text bigger and improve quality of figure – use another software
perhaps.
Response: Thank you for the suggestion. This figure has been modified (Figure 5).
- Figure 6 – the police in between your figures are all different, the resolution is still low - This
does not constitute an answer to my first review and does not respond to is properly.
Response: Thank you for the suggestion. This picture has been modified (Figure 6).
- Figure 7 - This does not constitute an answer to my first review and does not respond to is properly. – still unreadable.
Response: Thanks to your suggestion, we shortened the spacing of the horizontal coordinate data and re-made the figure 7.
- L.335-340 The results do not support your conclusion – you didn’t prove that E. onukii can
discriminate red/yellow patterns from yellow only from a distance.
Response: We apologize for any confusion. We have modified this expression.
‘According to the simulated visual acuity of 0.14 cpd, adult E. onukii could perceive blurred details of a 20×25 cm picture through vision at a distance of 0–30 cm (Figure 8B–D). Beyond a distance of 40 cm, differences within the picture could not be distinguished (Figure 8E). The results showed that the E. onukii compound eyes had limited ability to distinguish the details of the image and could only recognize a large object within a certain distance, but the resolution was low. The recognition distance was related to the object volume (line 290-295).’
- L. 374 ‘long range.’ Maybe even short range! You didn’t show the opposite. Behavioural experiments using the two sticky card types with different range have to be added to support your claims.
Response: We apologize for any confusion. We have rewritten this whole paragraph, and this sentence has been deleted. In the revised manuscript, we deemphasized the content describing the ability of E. onukii to distinguish colors and focused on the visual acuity of the leafhopper. In the section of discussion, we have added your suggestion as ‘the reason why bicolor sticky and yellow sticky cards showed similar trapping effects in the field experiment needs further research, including the effective attraction range of two types of sticky cards and flight ability of E. onukii (line 405-407).’
